# Multimodal prehabilitation (Fit4Surgery) in high-impact surgery to enhance surgical outcomes: Study protocol of F4S PREHAB, a single center stepped wedge trial

**Dieuwke Strijker[1]**, **Luuk Drager** [2]*, **Monique van Asseldonk[3]**, **Femke Atsma[4]**,
**Manon van den Berg[3]**, **Elke van Daal[2]**, **Linda van Heusden-Scholtalbers[5]**,
**Jeroen Meijerink[2]**, **Petra Servaes[6]**, **Steven Teerenstra[7]**, **Sjors Verlaan[2]**, **Baukje van den
Heuvel[2]**, **Kees van Laarhoven[1]**

1 Department of Surgery, Radboud University Medical Center, Nijmegen, the Netherlands, 2 Department of
Operating Rooms, Radboud University Medical Center, Nijmegen, the Netherlands, 3 Department of
Gastroenterology and Hepatology, Dietetics and Intestinal Failure, Radboud University Medical Center,
Nijmegen, the Netherlands, 4 Department of Health Sciences, IQ Healthcare, Radboud University Medical
Center, Nijmegen, the Netherlands, 5 Department of Rehabilitation, Radboud University Medical Center,
Nijmegen, the Netherlands, 6 Department of Medical Psychology, Radboud University Medical Center,
Nijmegen, the Netherlands, 7 Department of Health Evidence, Radboud University Medical Center,
Nijmegen, the Netherlands

☯ These authors contributed equally to this work.
* luuk.drager@radboudumc.nl

Raffaele, ITALY

**Data Availability Statement:** No datasets were
generated or analysed during the current study. All

## Abstract

### Background

High-impact surgery imposes a significant physiological and functional burden and is asso-
ciated with substantial postoperative morbidity. Multimodal prehabilitation has demonstrated
a reduction in postoperative complications and enhanced functional recovery, mainly in
abdominal cancer surgery. Common preoperative risk factors shared among patients
undergoing high-impact surgery, extending beyond abdominal cancer surgery procedures,
suggest the relevance of multimodal prehabilitation to a broader patient population. This
stepped wedge trial primarily aims to examine the hospital-wide effect of multimodal preha-
bilitation, compared to standard preoperative care, on the occurrence and severity of post-
operative complications. Secondary and tertiary endpoints include length of hospital stay,
physical fitness, nutritional status, mental health, intoxications, and cost-effectiveness of the
intervention.

### Methods

The Fit4Surgery (F4S) PREHAB trial is a monocenter stepped wedge trial in an academic
hospital. Adult patients, divided into 20 health clusters based on specific diagnoses, will be
assessed for eligibility and receive usual preoperative care or multimodal prehabilitation.
Patient enrollment commenced in March 2021 and continues up to and including April 2024.
The intervention consists of a high-intensity exercise program, a nutritional intervention,

relevant data from this study will be made available upon study completion.

**Funding:** Whey protein supplements (Nutri WheyTM Isolate) were provided by FrieslandCampina. https://www.frieslandcampina. com/nl/ FrieslandCampina did not play any role in the study design, data collection, analysis, decision to publish or preparation of the manuscript.

**Competing interests:** The authors have declared that no competing interests exist.

psychological support, and smoking and alcohol cessation. The primary outcome will be measured by the Clavien-Dindo classification (grade II or higher) and the Comprehensive Complication Index (CCI).

## Discussion

Multimodal prehabilitation potentially reduces postoperative complications and enhances functional recovery. This is the first study to determine the hospital-wide effect and cost-effectiveness of multimodal prehabilitation in patients across various surgical specialties.

## Background

Across various medical disciplines surgery continues to play a pivotal role as the cornerstone of treatment regimens. However, a substantial proportion of patients undergoing surgery experiences postoperative complications, which have consistently been associated with adverse outcomes including higher mortality rates, prolonged hospital stay, increased healthcare costs, and a diminished reported quality of life [1–4]. Moreover, surgery imposes a significant physiological and functional burden on patients, even in the absence of complications [5, 6].

In recent years, significant progress has been made in perioperative care to enhance postoperative outcomes and reduce morbidity rates. Notable advancements include the implementation of technical innovations such as minimal invasive surgery, the adoption of perioperative measures through the early recovery after surgery (ERAS) pathway, and centralization of high-risk cancer surgery leading to increased hospital volumes [7]. Nowadays, there is growing recognition of the importance to target the preoperative period, since patients' functional capacity has demonstrated consistent associations with postoperative outcomes [8, 9]. This change in emphasis has led to the emergence of prehabilitation, a process that aims to optimize modifiable preoperative risk factors to fortify patients' resilience against the demands of surgery [10].

Previously, prehabilitation studies focused on single modal interventions while current evidence suggests that multimodal prehabilitation programs yield more favorable effects on functional capacity [11, 12]. These comprehensive programs encompass various components such as physical exercise, nutritional counseling and protein supplementation, psychological support, and interventions targeting intoxicating habits like smoking and alcohol consumption [13].

Multimodal prehabilitation has been investigated in different surgical specialties, including abdominal cancer surgery, resulting in a reduction of postoperative complications and a shortened duration of hospital stay [14–17]. The shared focus on targeting common preoperative risk factors through prehabilitation suggests its potential applicability to other surgical specialties involving high-impact procedures, ultimately extending the benefits of improved postoperative outcomes to a larger patient population.

This stepped wedge trial primarily aims to examine the hospital-wide effect of a multimodal prehabilitation program compared to standard preoperative care on the occurrence and severity of postoperative complications in patients undergoing high-impact surgery across various surgical specialties. Furthermore, the effects of the intervention on length of hospital stay, physical fitness, nutritional status, mental health, and intoxications will be assessed and its cost-effectiveness will be evaluated.

## Methods

### Study design

This study is a monocenter stepped wedge trial at the Radboudumc in Nijmegen, the Netherlands. Ethical approval for this study was granted by the Institutional Review Board of the Radboudumc and local Medical Ethics Committee (METC Oost-Nederland) (NL73777.091.20) (S1 and S2 Files). The trial has been registered in the International Clinical Trials Registry Platform (NL8699). Any proposed amendments to the study protocol were submitted to the relevant Medical Ethics Committee for review and approval. Following the stepped wedge approach, a multimodal prehabilitation program will be implemented as standard preoperative care in one cluster every month. The order of clusters is determined based on logistical feasibility within the participating surgical departments. Patient enrollment and prospective data collection commenced on 1 March 2021 and is planned to continue up to and including 30 April 2024. Additionally, retrospective data regarding the primary outcomes pertaining to patients who underwent elective high-impact surgery during the two-year period preceding the initiation of this trial will be gathered. A SPIRIT schedule and overview of the study design can be found in Figs 1 and 2 (S1 Checklist).

### Study population

All patients aged sixteen years and older undergoing elective high-impact surgery within 20 health clusters will be assessed for study eligibility by both the medical specialist and investigator. During the initial outpatient clinic visit, patients will be provided with detailed

|  | STUDY PERIOD | | | | | | |
|---|---|---|---|---|---|---|---|
|  | Enrolment | Allocation | Post-allocation | | | | Close-out |
| **TIMEPOINT** | *-t1* | **0** | *t1* | *t2* | *t3* | *t4* | *t5* |
| **ENROLMENT:** | | | Baseline | After prehabilitation | +3 months after surgery | +6 months after surgery | +12 months after surgery |
| Eligibility | X | | | | | | |
| Informed consent | X | | | | | | |
| Allocation | | X | | | | | |
| **INTERVENTIONS:** | | | | | | | |
| Exercise program | | | ◆————————◆ | | | | |
| Nutritional intervention | | | ◆————————◆ | | | | |
| Psychological support | | | ◆————————◆ | | | | |
| Intoxication cessation | | | ◆————————◆ | | | | |
| **ASSESSMENTS:** | | | | | | | |
| Functional capacity and activity | | | X | X | X | | |
| Nutritional status | | | X | X | X | | |
| Mental health | | | X | X | X | X | X |
| Medical consumption | | | | X | X | X | X |
| Health behavior | | | | X | X | | |

**Fig 1. SPIRIT schedule of enrolment, interventions, and assessments for the F4S PREHAB study.**

| Cluster | Before Implementation (24 months) | | | | Period of stepped wedge implementation (20 months) | | | | | | | | | | | | | | | | | | | | After implementation | | | |
|---|---|---|---|---|---|---|---|---|---|---|---|---|---|---|---|---|---|---|---|---|---|---|---|---|---|---|---|---|---|
| 1 | 0 | ... | ... | 0 | 1 | 1 | 1 | 1 | 1 | 1 | 1 | 1 | 1 | 1 | 1 | 1 | 1 | 1 | 1 | 1 | 1 | 1 | 1 | 1 | 1 | 1 | 1 | 1 |
| 2 | 0 | ... | ... | 0 | 0 | 1 | 1 | 1 | 1 | 1 | 1 | 1 | 1 | 1 | 1 | 1 | 1 | 1 | 1 | 1 | 1 | 1 | 1 | 1 | 1 | 1 | 1 | 1 |
| 3 | 0 | ... | ... | 0 | 0 | 0 | 1 | 1 | 1 | 1 | 1 | 1 | 1 | 1 | 1 | 1 | 1 | 1 | 1 | 1 | 1 | 1 | 1 | 1 | 1 | 1 | 1 | 1 |
| 4 | 0 | ... | ... | 0 | 0 | 0 | 0 | 1 | 1 | 1 | 1 | 1 | 1 | 1 | 1 | 1 | 1 | 1 | 1 | 1 | 1 | 1 | 1 | 1 | 1 | 1 | 1 | 1 |
| 5 | 0 | ... | ... | 0 | 0 | 0 | 0 | 0 | 1 | 1 | 1 | 1 | 1 | 1 | 1 | 1 | 1 | 1 | 1 | 1 | 1 | 1 | 1 | 1 | 1 | 1 | 1 | 1 |
| 6 | 0 | ... | ... | 0 | 0 | 0 | 0 | 0 | 0 | 1 | 1 | 1 | 1 | 1 | 1 | 1 | 1 | 1 | 1 | 1 | 1 | 1 | 1 | 1 | 1 | 1 | 1 | 1 |
| 7 | 0 | ... | ... | 0 | 0 | 0 | 0 | 0 | 0 | 0 | 1 | 1 | 1 | 1 | 1 | 1 | 1 | 1 | 1 | 1 | 1 | 1 | 1 | 1 | 1 | 1 | 1 | 1 |
| 8 | 0 | ... | ... | 0 | 0 | 0 | 0 | 0 | 0 | 0 | 0 | 1 | 1 | 1 | 1 | 1 | 1 | 1 | 1 | 1 | 1 | 1 | 1 | 1 | 1 | 1 | 1 | 1 |
| 9 | 0 | ... | ... | 0 | 0 | 0 | 0 | 0 | 0 | 0 | 0 | 0 | 1 | 1 | 1 | 1 | 1 | 1 | 1 | 1 | 1 | 1 | 1 | 1 | 1 | 1 | 1 | 1 |
| 10 | 0 | ... | ... | 0 | 0 | 0 | 0 | 0 | 0 | 0 | 0 | 0 | 0 | 1 | 1 | 1 | 1 | 1 | 1 | 1 | 1 | 1 | 1 | 1 | 1 | 1 | 1 | 1 |
| 11 | 0 | ... | ... | 0 | 0 | 0 | 0 | 0 | 0 | 0 | 0 | 0 | 0 | 0 | 1 | 1 | 1 | 1 | 1 | 1 | 1 | 1 | 1 | 1 | 1 | 1 | 1 | 1 |
| 12 | 0 | ... | ... | 0 | 0 | 0 | 0 | 0 | 0 | 0 | 0 | 0 | 0 | 0 | 0 | 1 | 1 | 1 | 1 | 1 | 1 | 1 | 1 | 1 | 1 | 1 | 1 | 1 |
| 13 | 0 | ... | ... | 0 | 0 | 0 | 0 | 0 | 0 | 0 | 0 | 0 | 0 | 0 | 0 | 0 | 1 | 1 | 1 | 1 | 1 | 1 | 1 | 1 | 1 | 1 | 1 | 1 |
| 14 | 0 | ... | ... | 0 | 0 | 0 | 0 | 0 | 0 | 0 | 0 | 0 | 0 | 0 | 0 | 0 | 0 | 1 | 1 | 1 | 1 | 1 | 1 | 1 | 1 | 1 | 1 | 1 |
| 15 | 0 | ... | ... | 0 | 0 | 0 | 0 | 0 | 0 | 0 | 0 | 0 | 0 | 0 | 0 | 0 | 0 | 0 | 1 | 1 | 1 | 1 | 1 | 1 | 1 | 1 | 1 | 1 |
| 16 | 0 | ... | ... | 0 | 0 | 0 | 0 | 0 | 0 | 0 | 0 | 0 | 0 | 0 | 0 | 0 | 0 | 0 | 0 | 1 | 1 | 1 | 1 | 1 | 1 | 1 | 1 | 1 |
| 17 | 0 | ... | ... | 0 | 0 | 0 | 0 | 0 | 0 | 0 | 0 | 0 | 0 | 0 | 0 | 0 | 0 | 0 | 0 | 0 | 1 | 1 | 1 | 1 | 1 | 1 | 1 | 1 |
| 18 | 0 | ... | ... | 0 | 0 | 0 | 0 | 0 | 0 | 0 | 0 | 0 | 0 | 0 | 0 | 0 | 0 | 0 | 0 | 0 | 0 | 1 | 1 | 1 | 1 | 1 | 1 | 1 |
| 19 | 0 | ... | ... | 0 | 0 | 0 | 0 | 0 | 0 | 0 | 0 | 0 | 0 | 0 | 0 | 0 | 0 | 0 | 0 | 0 | 0 | 0 | 1 | 1 | 1 | 1 | 1 | 1 |
| 20 | 0 | ... | ... | 0 | 0 | 0 | 0 | 0 | 0 | 0 | 0 | 0 | 0 | 0 | 0 | 0 | 0 | 0 | 0 | 0 | 0 | 0 | 0 | 1 | 1 | 1 | 1 | 1 |

**Fig 2. Schematic of the study design.** A '0' indicates that patients in the corresponding cluster receive standard preoperative care. A '1' indicates that patients in the corresponding cluster undergo the multimodal prehabilitation program.

information regarding the study. Prior to participation, written informed consent will be obtained from all patients. Patients will be divided into 20 clusters based on their specific diagnoses, including colon cancer, rectal cancer, liver cancer or metastases (of colorectal origin), (retro)peritoneal malignancies, esophageal cancer, pancreaticobiliary cancer, abdominal aortic aneurysm (open and endovascular repair), oral cancer, laryngeal cancer, supratentorial meningioma, autologous breast reconstruction, endometrial cancer, ovarian cancer, vulvar cancer, hip arthrosis, hip or knee arthroplasty failure, renal cancer, and bladder cancer (Table 1).

Exclusion criteria comprise chronic kidney disease stage ≥4 which contraindicates protein supplementation, cognitive disabilities or illiteracy (inability to read and understand the Dutch language), and characteristics that contraindicate or impede high-intensity exercise such as impaired mobility, premorbid conditions like cardiac or respiratory diseases, and ASA scores of ≥4.

## Sample size

The sample size calculation was based on simulations in a log-binomial model with cluster (surgical procedure), intervention, and time (as continuous variable) as fixed effects. A log-binomial model was used in order to be able to calculate risk ratios, since odds ratios

**Table 1. Estimated surgical procedures per year and overall registered complication rates in the Radboudumc.**

| # | Health clusters | Numbers of surgical procedures/year | Overall complication rates (Clavien-Dindo II or higher) |
|---|---|---|---|
| 1 | Colon cancer | 65 | 15% |
| 2 | Rectal cancer | 60 | 15% |
| 3 | Liver cancer or metastases (CRC origin) | 100 | 15% |
| 4 | (Retro)peritoneal malignancies | 60 | 30% |
| 5 | Esophageal cancer | 80 | 55% |
| 6 | Pancreaticobiliary cancer | 78 | 33% |
| 7 | Abdominal aortic aneurysm (open repair) | 35 | 58% |
| 8 | Abdominal aortic aneurysm (endovascular repair) | 120 | 24% |
| 9 | Oral cancer (free flap reconstruction) | 60 | 35%* |
| 10 | Laryngeal cancer | 65 | 30%* |
| 11 | Supratentorial meningeoma | 55 | 11% |
| 12 | Autologous breast reconstruction | 118 | 22% |
| 13 | Endometrial cancer | 40 | 10% |
| 14 | Ovarian cancer | 80 | 30% |
| 15 | Vulvar field resection | 40 | 18% |
| 16 | Hip arthrosis | 50 | 15%* |
| 17 | Hip arthroplasty failure | 70 | 20%* |
| 18 | Knee arthroplasty failure | 50 | 15%* |
| 19 | Renal cancer | 60 | 25% |
| 20 | Bladder cancer | 78 | 38% |

* Percentages based on literature [18–21].

overestimate risk in case of high prevalent outcomes [22]. Fixed effects for clusters are chosen as the aim is primarily to establish the effect for the Radboudumc. Monthly binary data were generated per cluster (surgical procedure) per month with a binary distribution according to the sample size corresponding to known patient numbers and complication rates in the Radboudumc. The power calculation assumes a relative reduction of 20% in complication rates (Clavien-Dindo II or higher). Based on these simulations and the expected recruitment, the power was 83%. The total prospective recruitment in terms of patients was expected to be 2828 (2 years with 1414 patients per year). The prospective cohort will be complemented by a 2-year historical cohort, consisting of an expected number of 2828 patients as well.

## Study outline

Patients participating in the control group will receive standard preoperative care in accordance with Dutch guidelines. Patients assigned to the intervention group will undergo a personalized multimodal prehabilitation program, which will span three to four weeks depending on the available time between diagnosis and surgery. The program comprises four distinct components: an exercise program, a nutritional intervention, psychological support, and smoking and alcohol cessation. Hemoglobin and glucose levels will also be assessed at the start of the program and will be addressed and treated accordingly.

Additionally, for study purposes and data collection, all patients in the prospective period will attend three additional appointments alongside their regular preoperative and postoperative outpatient clinic visits. These appointments will be arranged prior to the control period or the commencement of the prehabilitation program, shortly before surgery, and three months postoperatively. Data regarding physical fitness, nutritional status, mental health, and health

**Table 2. Overview of preoperative, perioperative, and postoperative measurements.**

| Baseline | After prehabilitation and prior to surgery | Perioperative period | +3 months | +6 months | +12 months |
|---|---|---|---|---|---|
| *Functional capacity and activity* VO$_2$ peak (Steep Ramp Test) • Indirect 1 repetition measures (1RM) • Hand grip strength • Timed Chair Stand test (5-CST) • Short Questionnaire to Assess Health-enhancing physical activity (SQUASH) | *Functional capacity and activity* VO$_2$ peak (Steep Ramp Test) • Indirect 1 repetition measures (1RM) • Hand grip strength • Timed Chair Stand test (5-CST) | *Postoperative complications* • Clavien-Dindo classification • Comprehensive Complication Index (CCI) | *Functional capacity and activity* VO$_2$ peak (Steep Ramp Test) • Indirect 1 repetition measures (1RM) • Hand grip strength • Timed Chair Stand test (5-CST) Short Questionnaire to Assess Health-enhancing physical activity (SQUASH) | *Functional activity* • Short Questionnaire to Assess Health-enhancing physical activity (SQUASH) | *Functional activity* • Short Questionnaire to Assess Health-enhancing physical activity (SQUASH) |
| *Nutritional status* • Length • Body weight (BW) • Fat-free mass (FFM) • PG-SGA SF questionnaire • 3-day food diary | *Nutritional status* • Length • Body weight (BW) • Fat-free mass (FFM) • PG-SGA SF questionnaire • 3-day food diary | *Length of hospital stay (LoS)* | *Nutritional status* • Length • Body weight (BW) • Fat-free mass (FFM) • PG-SGA SF questionnaire | *Nutritional status* N/A | *Nutritional status* N/A |
| *Mental health* • SF-36 questionnaire • Hospital Anxiety and Depression Scale (HADS) | *Mental health* • SF-36 questionnaire | | *Mental health* • SF-36 questionnaire | *Mental health* • SF-36 questionnaire • EuroQuol 5D (EQ-5D-5L) | *Mental health* • SF-36 questionnaire |
| *Medical consumption* • N/A | *Medical consumption* • iMedical Consumption Questionnaire (iMCQ) | | *Medical consumption* • iMedical Consumption Questionnaire (iMCQ) | *Medical consumption* • iMedical Consumption Questionnaire (iMCQ) | *Medical consumption* • iMedical Consumption Questionnaire (iMCQ) |
| *Health behavior* • N/A | *Health behavior* • Health behavior questionnaire | | *Health behavior* • Health behavior questionnaire | | |

behavior in both the pre- and postoperative phases will be collected (Table 2). Retrospectively included patients will only be assessed for the primary and secondary outcomes.

**Physical assessment and exercise program.** Prior to the assessment and exercise program, patients will undergo screening using the American College of Sports Medicine (ACSM) exercise preparticipation health screening questionnaire to identify individuals who may have an elevated risk of exercise-related sudden cardiac death or acute myocardial infarction [23]. Patients identified as being at risk will receive an additional referral to a cardiologist or pulmonologist to evaluate the safety of engaging in high intensity interval training (HIIT).

Patients' physical capacity and mobility will be evaluated by an in-hospital physiotherapist. The content of the exercise program will be customized based on the estimated maximal oxygen consumption (VO$_2$ peak) and indirect one repetition maximum (1RM) measured at baseline, utilizing a Steep Ramp Test and leg press, respectively. All exercise training sessions will be supervised by a physiotherapist in a primary healthcare setting and will be conducted three times a week. The goal of the exercise program is to achieve a 10% increase in estimated VO$_2$ peak and indirect 1RM measurements.

The exercise program consists of the following components:

- Endurance training (HIIT):

- Interval training with a total duration of 28 minutes, preceded by a 2-minute warm-up, consisting of alternating intervals of high intensity (4 intervals of 4 minutes) and moderate intensity (4 intervals of 3 minutes).

- The high intensity workload will be set at 90% of the peak wattage achieved in the Steep Ramp Test, corresponding to an estimated 90% $VO_2$ peak. The aim is to achieve Borg scores of 15–17 and $\geq$ 85% of the age-predicted maximal heart rate [24, 25].

- The moderate intensity workload will be set at 30% of the peak wattage.

- Examples of aerobic exercise machines suitable for performing HIIT include a bicycle, rower, treadmill, and cross-trainer.

- The workload should be adjusted by 5–10% if a patient is unable to complete the high intensity intervals.

- Resistance training:

- This training targets all major muscle groups and comprises six exercises: leg press, chest press, abdominal crunch, low row, lat pulldown, and step up. Each exercise consists of two sets of 10 repetitions.

- The strength exercises will follow a pattern of two seconds of concentric strength and two seconds of eccentric strength. The weight for each exercise will be adjusted based on the indirect 1RM measured at baseline [26]. The weight for the exercises will start at 65% of the calculated 1RM, with a weekly increase of 5% resulting in 80% of baseline 1RM by the fourth week of the exercise program.

- The weight should be adjusted by 5–10% based on a patient's ability to complete 10 repetitions in the second set.

- Unsupervised training:

- Patients will be instructed to engage in at least 60 minutes of aerobic exercise on days without supervised training. If physical capacity is insufficient, this can be divided into two or three periods of 20–30 minutes. Examples of aerobic exercises include walking, cycling, and swimming.

- All patients will be provided with guidance on adequate rest and sleep.

**Nutritional assessment and intervention.** Nutritional assessment will be conducted at baseline and prior to surgery, involving the collection of the following parameters: body weight (BW), length, fat-free mass measured by bioelectrical impedance analysis (BIA), and a 3-day food diary. The Patient-Generated Subjective Global Assessment (PG-SGA) will be used to provide a general impression of the patients' nutritional status.

Furthermore, all patients will be referred to a registered in-hospital dietician who will provide personalized dietary advice to optimize nutritional intake, focusing on protein, energy, and micronutrients. This guidance aims to achieve an anabolic state and to enhance the effects of physical training on lean body mass increment. Since protein intake is crucial to stimulate muscle protein synthesis, the goal is to achieve a daily protein intake of $\geq$1.5 g/kg/BW and a minimal intake of $\geq$1.2 g/kg/BW [27, 28]. To accomplish this, patients will receive high-quality whey protein shakes (Nutri Whey™ Isolate, FrieslandCampina) containing 30 g of whey protein and 20 μg vitamin D to be consumed as one dose daily and an additional dose following supervised training (within one hour). Patients will be advised to distribute their dietary

protein consumption evenly across meals, aiming for at least two meals per day containing 25 g protein or more.

To address potential vitamin deficiencies all patients will be provided with daily multivitamin supplementation, equivalent to 50% of the recommended daily intake.

**Psychological counseling and support.** To assess patients' symptoms of anxiety and depression the Hospital Anxiety and Depression Scale (HADS) will be applied at baseline [29]. Patients with scores ≥15 on the HADS will be referred to a trained psychologist who will provide support to optimize their psychological well-being and teach coping mechanisms specifically tailored to the surgical treatment. Patients with scores lower than 15 and a history of psychological health issues will be referred to a social worker. Additional sessions will be scheduled during the preoperative period as deemed necessary.

**Smoking and alcohol cessation.** All patients who are active smokers during the baseline assessment will be offered a comprehensive smoking cessation program, which includes counseling and nicotine replacement therapy, prior to surgery. Furthermore, all patients are advised to completely quit the consumption of alcoholic beverages.

**Postoperative period.** Patients in both the control and intervention group will receive standard postoperative care in accordance with Dutch guidelines.

## Study outcomes

The primary outcomes of this study will be the occurrence and severity of postoperative complications, as measured by the Clavien-Dindo classification and Comprehensive Complication Index (CCI). Complications will be assessed 30 days after surgery, utilizing medical data documented in electronic patient files (EPF). Clavien-Dindo scores of II or higher will be used to indicate the difference in risk of postoperative complications between the control and intervention group. Additionally, CCI scores will be calculated based on the sum of complications according to the Clavien-Dindo classification to assess complication severity [30–32].

Secondary study outcome reflects the length of hospital stay (LoS), which will be extracted from the EPF as well. Tertiary parameters include functional capacity (estimated VO$_2$ peak, indirect 1RM, handgrip strength, Timed Chair Stand test, SQUASH questionnaire), nutritional status (length, BW, fat-free mass, PG-SGA questionnaire), mental health status (SF-36 and EQ-5D-5L questionnaires), intoxications (Health Behavior Questionnaire), and medical consumption (iMedical Consumption Questionnaire). Tertiary outcomes will be measured at baseline, prior to surgery, and three months postoperatively (within a range of 10–14 weeks). Potential differences in tertiary outcomes between control and intervention group will be assessed prior to surgery, as well as three months post-surgery. Additionally, follow-up questionnaires for some tertiary parameters will be sent at 6 and 12 months post-surgery, allowing for potential additional analyses to be performed. Other tertiary outcomes to be evaluated comprise adherence to the intervention and cost-effectiveness. Table 2 shows an overview regarding the measurements and secondary and tertiary outcomes. Subgroup analyses will be conducted based on training volume (defined as ≥9 sessions or 80% of the prescribed number of sessions), socioeconomic status (SES), patients undergoing abdominal cancer surgery, and other relevant determinants.

## Process evaluation

Facilitators and barriers impacting the process of implementation will be assessed by a process evaluation. Furthermore, this evaluation aims to delve into the contextual elements of significance and to capture the firsthand experiences of patients and healthcare practitioners concerning the specific components of the multimodal prehabilitation program. Achieving this

will entail organizing interviews or focus groups that involve both patients and healthcare providers.

## Statistical analysis

Baseline descriptive characteristics of the patient cohort will be described as mean and standard deviation (SD) for continuous variables, and count and percentage for categorical variables. The effect of the intervention on the primary outcome, the risk of Clavien-Dindo grade II complications or higher, will be analyzed using a log-binomial model (generalized linear model with log-link and binomial error distribution) fitted with clusters (categorical variable), treatment arm, and time (as linear continuous) as fixed effects. A log-binomial regression will be employed to compute risk ratios instead of odds ratios, as the primary outcome is expected to be relatively prevalent. Odds ratios tend to overestimate risk ratios in such scenarios [22]. Fixed effects for clusters are chosen as the aim is primarily to establish the effect for the Radboudumc. The linearity assumption for time will be assessed by including quadratic terms or other polynomials. Additionally, CCI scores will be analyzed using generalized linear models with log link, normal distribution for the errors, and the same fixed effects. Due to the potential dependency of data within clusters, linear mixed effects models incorporating random intercepts and/or slopes for clusters, will be conducted and compared to the previously mentioned models. Besides, model assumptions will be checked and in case of violated assumptions, other parametrizations will be used. The Aikake information criteria (AIC) will be employed for model selection models within the same class. Primary analyses will be performed according to an intention-to-treat as well as a per-protocol approach.

Secondary outcome (LoS) will be analyzed using a linear regression model. Similarly to the primary outcomes, assumptions for linear regression will be tested and in case of violation, alternative distributions will be modelled. Regarding the tertiary parameters, the potential differences in outcomes between control and intervention group will be tested prior to surgery, as well as three months post-surgery. Independent samples t-tests and Mann-Whitney U tests will be applied to analyze normally and non-normally distributed continuous parameters, whereas chi-square tests or logistic, ordinal or nominal regression models will be used to analyze categorical parameters.

## Economic evaluation

The cost-effectiveness analysis will be conducted from a healthcare perspective. It aims to measure, value, and analyze healthcare costs at a patient level for those undergoing usual preoperative care and those undergoing a multimodal prehabilitation program. The time horizon of the economic evaluation is 12 months after surgery, and will be performed alongside the stepped wedge trial. To expand the efficiency analysis, medical consumption will be measured by the iMedical Consumption Questionnaire (iMCQ) shortly before surgery and 3, 6, and 12 months after surgery.

Standard cost prices for postoperative complications and prehabilitation programs will be determined based on the 'Kostenhandleiding' [33]. A multi-level generalized linear model with gamma distribution and log-link function (if data are non-normally distributed) will be used for the cost analysis, with fixed effects for time and random effects for clusters [34].

Additionally, a cost-utility analysis will be performed with quality-adjusted life-years (QALYs) based on the EQ-5D-5L at 6 months after surgery, following the Dutch guidelines for economic evaluation in healthcare [35].

## Discussion

High-impact surgery is associated with substantial postoperative morbidity, significantly affecting patients' physiological and functional capacity [6, 36, 37]. Multimodal prehabilitation has demonstrated promising potential to improve postoperative outcomes. However, the current evidence is limited due to the presence of small sample size studies and high heterogeneity in prehabilitation programs. Moreover, there has been a predominant emphasis on assessing the impact of multimodal prehabilitation in patients undergoing abdominal cancer surgery. To overcome these limitations, we have initiated a monocenter stepped wedge trial to assess the hospital-wide effect of a uniform multimodal prehabilitation program on the occurrence and severity of postoperative complications among patients undergoing elective high-impact surgery across a wide range of surgical fields. In addition, this study will assess the impact of multimodal prehabilitation on length of hospital stay, physical fitness, nutritional status, mental health, and intoxications, and will determine its cost-effectiveness.

This stepped wedge trial will include patients from 20 different health clusters. In light of prior studies showcasing a notable reduction of up to 50% in postoperative complications following prehabilitation in colorectal cancer and major abdominal surgery, the potential of prehabilitation is highly promising [14, 15, 38]. These encouraging findings suggest a need to explore whether the merits of multimodal prehabilitation might extend to a wider range of patients undergoing different types of high-impact surgery, possessing similar modifiable preoperative risk factors addressed by these interventions. The selection of distinct health clusters was based on reported postoperative morbidity levels, aiming to include patients who could potentially benefit the most from the intervention. An additional consideration was the available timeframe during the preoperative period within these health clusters. Although the ideal timeframe for prehabilitation has not been established, past investigations have indicated positive effects within just three weeks [14, 39, 40]. Consequently, to ensure effective integration of multimodal prehabilitation, health clusters with a minimum preoperative period of three weeks were included. Health clusters involved in ongoing or planned scientific research, possibly interfering with multimodal prehabilitation, were excluded to preserve the integrity of the implementation process.

Assessing the effectiveness of multimodal prehabilitation across numerous health clusters in a hospital-wide context necessitates a pragmatic approach. To ensure systematic and comprehensive implementation in each health cluster, a stepped wedge design was chosen, which is well-suited for evaluating interventions during their implementation into routine practice [41]. The design choice not only aligns with pragmatism but also addresses ethical considerations. Some recent studies investigating multimodal prehabilitation have utilized randomized parallel-arm controlled trials to examine its impact on postoperative complications [14, 15, 42]. The trial conducted by Berkel et al. highlighted the challenges associated with patient inclusion [14]. While not explicitly stated, randomization, and the consequent withholding of a potentially beneficial intervention from a particular group of patients, could have hindered patient inclusion due to ethical concerns from both patient and healthcare perspectives. Moreover, incorporation of multimodal prehabilitation into standard preoperative care emphasizes the importance of potential health benefits, possibly leading to both enhanced participation rates and increased patient motivation.

The F4S PREHAB trial has opted for a single-center approach to evaluate the hospital-wide effects in the Radboudumc. Due to pending scientific evidence on the effectiveness of prehabilitation, there is an absence of established financial options within the Dutch healthcare system. Consequently, the Radboudumc, as an early adopter, has taken full financial responsibility for all trial-related investments. This financial commitment allows for hospital-

wide outcomes being of utmost relevance to other healthcare institutions contemplating the integration of multimodal prehabilitation as a standard preoperative care practice across various surgical pathways beyond abdominal cancer surgery.

The study protocol has undergone some noteworthy modifications from the original plan. Firstly, the order of health clusters was intended to be randomized as opposed to the current non-randomized approach. Considering the scale and complexity of hospital-wide implementation, logistic feasibility was considered most important in this pragmatic trial to enable successful implementation across all health clusters. Secondly, the initial 24-months duration for patient recruitment was extended. The sample size and power calculations were based on historical surgical procedure volumes at the Radboudumc. However, due to a lower-than-expected number of surgical procedures, worsened by the impact of the COVID-19 pandemic, the study duration was prolonged up to and including April 2024.

While hospital-wide implementation of multimodal prehabilitation seems pivotal to enhance postoperative outcomes, this stepped wedge trial does have limitations. First of all, the sample size and power estimation are based on a relative reduction in overall complication rates at a hospital level. Consequently, it may not be possible to determine statistically significant differences in the primary outcome for individual health clusters. Secondly, there is a risk of performance bias as both the usual care and prehabilitation groups are informed about the potential benefits gained by multimodal prehabilitation. To mitigate this risk, patients in the usual care group are not provided with detailed information regarding the specific contents of the intervention.

In conclusion, this study aims to provide insight into the hospital-wide effects of implementing a multimodal prehabilitation program on clinical and functional outcomes in a variety of patients undergoing high-impact surgery. By assessing the effects on a hospital-wide level, this study seeks to provide valuable insights into the efficacy and potential benefits of implementing multimodal prehabilitation as a standard practice. Additionally, a detailed cost-effectiveness analysis will be conducted to assess the financial implications and potential health value generated by the program. Together, these outcomes will contribute to the understanding of the broader benefits and feasibility of multimodal prehabilitation in optimizing patient outcomes and healthcare resource utilization.

## Supporting information

**S1 Checklist. SPIRIT checklist.**
(DOC)

**S1 Protocol. Study protocol approved by the Ethics Committee.**
(PDF)

**S1 File. Proof of ethics approval Dutch.**
(PDF)

**S2 File. Proof of ethics approval English.**
(PDF)

## Acknowledgments

The authors thank dr. Eddy Adang of the department of Health Evidence Radboud University Medical Center for his support in writing this study protocol.

## Author Contributions

**Conceptualization:** Dieuwke Strijker, Luuk Drager, Monique van Asseldonk, Manon van den Berg, Elke van Daal, Linda van Heusden-Scholtalbers, Jeroen Meijerink, Petra Servaes, Baukje van den Heuvel, Kees van Laarhoven.

**Investigation:** Dieuwke Strijker, Luuk Drager.

**Methodology:** Dieuwke Strijker, Luuk Drager, Elke van Daal, Jeroen Meijerink, Steven Teerenstra, Baukje van den Heuvel.

**Project administration:** Dieuwke Strijker, Luuk Drager, Baukje van den Heuvel, Kees van Laarhoven.

**Resources:** Sjors Verlaan.

**Supervision:** Baukje van den Heuvel, Kees van Laarhoven.

**Visualization:** Dieuwke Strijker, Luuk Drager.

**Writing – original draft:** Dieuwke Strijker, Luuk Drager.

**Writing – review & editing:** Dieuwke Strijker, Luuk Drager, Monique van Asseldonk, Femke Atsma, Manon van den Berg, Elke van Daal, Linda van Heusden-Scholtalbers, Jeroen Meijerink, Petra Servaes, Steven Teerenstra, Sjors Verlaan, Baukje van den Heuvel, Kees van Laarhoven.

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
