## [Decision Letter · Decision Letter 0]

28 Feb 2024

PONE-D-24-01585Multimodal Prehabilitation (Fit4Surgery) in High-impact Surgery to Enhance Surgical Outcomes: Study Protocol of F4S PREHAB, a Single Center Stepped Wedge TrialPLOS ONE

Dear Dr. Drager,

Thank you for submitting your manuscript to PLOS ONE. After careful consideration, we feel that it has merit but does not fully meet PLOS ONE’s publication criteria as it currently stands. Therefore, we invite you to submit a revised version of the manuscript that addresses the points raised during the review process.

We look forward to receiving your revised manuscript.

Kind regards,

Stefano Turi

Academic Editor

PLOS ONE

Reviewers' comments:

Reviewer's Responses to Questions

**Comments to the Author**

1. Does the manuscript provide a valid rationale for the proposed study, with clearly identified and justified research questions?

Reviewer #1: Yes

Reviewer #2: Yes

2. Is the protocol technically sound and planned in a manner that will lead to a meaningful outcome and allow testing the stated hypotheses?

Reviewer #1: Yes

Reviewer #2: Partly

3. Is the methodology feasible and described in sufficient detail to allow the work to be replicable?

Reviewer #1: Yes

Reviewer #2: Yes

4. Have the authors described where all data underlying the findings will be made available when the study is complete?

Reviewer #1: Yes

Reviewer #2: Yes

5. Is the manuscript presented in an intelligible fashion and written in standard English?

Reviewer #1: Yes

Reviewer #2: Yes

6. Review Comments to the Author

You may also provide optional suggestions and comments to authors that they might find helpful in planning their study.

Reviewer #1: Study design is pragmatic rather than an RCT which would yield the highest level of evidence. The authors have argumented well why this is done.

It a well designed, clearly described protocol of an important project. I will be keen to see the results.

Reviewer #2: The manuscript addresses an interesting topic. The study is valuable and well described, though more details are required on several parts. Some comments are provided to ensure the reliability and reproducibility of the results.

1. Sample size. This Section should be rewritten and more details provided. As it stands, it is rather obscure. Please, justify why "a log-binomial model with fixed effects of cluster (specific diagnosis), intervention, and time trend

in absence of intervention (as a continuous variable)" is considered. It is rather unclear to me why a log-binomial model should be used in such a context. Similarly, time trend could be non-linear, did you consider the linear term only? Similarly, please clarify what you mean with "assuming a binary distribution that correlates with the estimated number of surgical procedures"; how do you compute correlations involving binary variables? I strongly suggest to write down the proper equations, along with all model's assumptions. The text, as it stands, is not informative to ensure the reproducibility of the method.

2. Statistical modelling. Again, it is rather unclear to me why a log-link is considered instead of the canonical logit link; please, make sure that you justify this in detail, aren't you interested in odds-ratios?. Similarly, it is rather weird to assume Gaussian-distributed error terms, as the log-binomial model assumes binomial errors. I strongly suggest to avoid the inclusion of cluster-specific fixed effects (are you using simple dummy variables?), but rather to consider random effects specifications (maybe with random slopes as well, as the effects of the independent variables may be cluster-specific). Please, make clear that the assumptions underlying the modelling will be properly checked to ensure the reliability of statistical inference.

3. Data structure is rather unclear to me. Do you have cross-section, longitudinal or survival-type data? If longitudinal, i.e. repeated measurements, data are available, this data feature must be properly modelled. This is fundamental, as "differences in outcomes between control and intervention group will be tested per single time point" could be misleading, as it ignores the data dynamics.

4. Economic evaluation. Please, justify why "A multi-level generalized linear model with gamma distribution and log-link function (if data are non-normally distributed) will be used for the cost analysis, with fixed effects for time and random effects for clusters" is considered (here random effects are correctly included in the linear predictor). Costs are known to be far from being Gaussian-distributed, as they are further defined on R+.

5. Goodness of fit of all the proposed models must be carefully checked (along with the underlying assumptions). Please, clarify how do you want to investigate this modelling feature.

7. PLOS authors have the option to publish the peer review history of their article (what does this mean?). If published, this will include your full peer review and any attached files.

Reviewer #1: No

Reviewer #2: No

---

## [Author Response · Author response to Decision Letter 0]

7 Mar 2024

Reviewer 1

Study design is pragmatic rather than an RCT which would yield the highest level of evidence. The authors have argumented well why this is done. It a well designed, clearly described protocol of an important project. I will be keen to see the results.

Your comment is greatly appreciated. Thank you for taking the time to review our manuscript. 

Reviewer 2

1. Sample size. This Section should be rewritten and more details provided. As it stands, it is rather obscure. Please, justify why "a log-binomial model with fixed effects of cluster (specific diagnosis), intervention, and time trend in absence of intervention (as a continuous variable)" is considered. It is rather unclear to me why a log-binomial model should be used in such a context. Similarly, time trend could be non-linear, did you consider the linear term only? Similarly, please clarify what you mean with "assuming a binary distribution that correlates with the estimated number of surgical procedures"; how do you compute correlations involving binary variables? I strongly suggest to write down the proper equations, along with all model's assumptions. The text, as it stands, is not informative to ensure the reproducibility of the method.

Thank you for your comment. We realize that our initial explanation might have been too brief. Our choice for a log-binomial model is based on the assumption that our intervention, multimodal prehabilitation, will lead to a relative reduction of 20% in complication rates, indicated by the occurrence of a Clavien-Dindo score II or higher. Given the binary nature of the outcome (complication vs. no complication) and our aim to compare relative risks between control and intervention group, we believe the log-binomial model used in our sample size simulation aligns with our research objectives.

You rightly point out the possibility that the time trend could be non-linear. We only assumed the time trend as a linear term in the power calculation as part of the simulation. However, of course, we will further explore this assumption in the statistical analyses of the collected data and will address non-linearity when needed. We changed the text of the manuscript accordingly.

2. Statistical modelling. Again, it is rather unclear to me why a log-link is considered instead of the canonical logit link; please, make sure that you justify this in detail, aren't you interested in odds-ratios?. Similarly, it is rather weird to assume Gaussian-distributed error terms, as the log-binomial model assumes binomial errors. I strongly suggest to avoid the inclusion of cluster-specific fixed effects (are you using simple dummy variables?), but rather to consider random effects specifications (maybe with random slopes as well, as the effects of the independent variables may be cluster-specific). Please, make clear that the assumptions underlying the modelling will be properly checked to ensure the reliability of statistical inference.

Our rationale for employing a log-binomial model stems from our objective to estimate risk ratios rather than odds ratios. One important consideration for this is that risk ratios are more accurate than odds ratios when the outcome is prevalent (1). Since we expect our primary outcome (the occurrence of Clavien-Dindo score II or higher) to be relatively common, we believe that using risk ratios for our analysis will be a valid approach. 

Regarding the modeling of error terms, there might be some confusion here. We indeed assume binomial-distributed error terms for our log-binomial model, instead of gaussian-distributed error terms. However, our second primary outcome, CCI scores (continuous outcome) requires a different approach and will be modeled using gaussian-distributed error terms. Since this may not have been clearly stated within our statistical analysis paragraph, we revised this in our new manuscript. 

We agree with the reviewer’s suggestion that our data may also be modelled using random effects for our surgical indications (clusters). Our suggested approach involves cluster-specific fixed effects, primarily driven as it aligns with our sample size calculation and focus on the Radboudumc effect. However, we acknowledge the importance of assessing the goodness-of-fit for our (proposed) models. Using mixed models is still an option, if this improves the model fit. To make this more clear, we have updated our manuscript to address more explicitly that the underlying assumptions of all our models will be tested. In case of violated assumptions, other parametrizations, such as random effects for clusters, will be used. 

3. Data structure is rather unclear to me. Do you have cross-section, longitudinal or survival-type data? If longitudinal, i.e. repeated measurements, data are available, this data feature must be properly modelled. This is fundamental, as "differences in outcomes between control and intervention group will be tested per single time point" could be misleading, as it ignores the data dynamics.

Thank you for your comment. Our tertiary data exhibits a longitudinal nature, as measurements are collected at three distinct time points: baseline, 1-2 days prior to surgery, and 3 months post-surgery. Our intention has been to test differences in tertiary outcomes prior to surgery between control and intervention group, as well as three months after surgery. Therefore, we state that “differences in outcomes between control and intervention group will be tested per single time point.” We acknowledge the need for further clarification on this aspect, which we provided in our revised manuscript. Moreover, since the analyses of our tertiary outcomes will be rather explorative, we may also explore the course of these outcomes over time. In case we will do that, the repeated measures in our data will be appropriately accounted for by using linear mixed-effects models.

4. Economic evaluation. Please, justify why "A multi-level generalized linear model with gamma distribution and log-link function (if data are non-normally distributed) will be used for the cost analysis, with fixed effects for time and random effects for clusters" is considered (here random effects are correctly included in the linear predictor). Costs are known to be far from being Gaussian-distributed, as they are further defined on R+.

When analyzing healthcare costs, which are often skewed, a generalized model with a gamma distribution is proposed as the most appropriate model (2). We added this reference in our manuscript to substantiate the use of this model distribution as first choice. As in the other analyses, we will also explore distributions and assumptions to ensure the model indeed gives the best fit in the cost analysis.

5. Goodness of fit of all the proposed models must be carefully checked (along with the underlying assumptions). Please, clarify how do you want to investigate this modelling feature.

As mentioned before, we underscore the importance of assessing the goodness-of-fit for our (proposed) models in order to ensure validity of statistical inference and choose the best models for our data. In our revised manuscript, we stated more clearly that model assumptions will be tested, and in case of violation, other parametrizations will be used. Moreover, to complement our assessment of goodness-of-fit, we will consult the Aikake information criteria (AIC) as a metric for model selection. 

References

1. Knol MJ, Le Cessie S, Algra A, Vandenbroucke JP, Groenwold RH. Overestimation of risk ratios by odds ratios in trials and cohort studies: alternatives to logistic regression. CMAJ. 2012;184(8):895-9.

2. Malehi AS, Pourmotahari F, Angali KA. Statistical models for the analysis of skewed healthcare cost data: a simulation study. Health Econ Rev. 2015;5:11.

---

## [Decision Letter · Decision Letter 1]

19 Mar 2024

PONE-D-24-01585R1Multimodal Prehabilitation (Fit4Surgery) in High-impact Surgery to Enhance Surgical Outcomes: Study Protocol of F4S PREHAB, a Single Center Stepped Wedge TrialPLOS ONE

Dear Dr. Drager,

Thank you for submitting your manuscript to PLOS ONE. After careful consideration, we feel that it has merit but does not fully meet PLOS ONE’s publication criteria as it currently stands. Therefore, we invite you to submit a revised version of the manuscript that addresses the points raised during the review process.

We look forward to receiving your revised manuscript.

Kind regards,

Stefano Turi

Academic Editor

PLOS ONE

Reviewers' comments:

Reviewer's Responses to Questions

**Comments to the Author**

1. Does the manuscript provide a valid rationale for the proposed study, with clearly identified and justified research questions?

Reviewer #1: Yes

Reviewer #2: Partly

2. Is the protocol technically sound and planned in a manner that will lead to a meaningful outcome and allow testing the stated hypotheses?

Reviewer #1: Yes

Reviewer #2: No

3. Is the methodology feasible and described in sufficient detail to allow the work to be replicable?

Reviewer #1: Yes

Reviewer #2: No

4. Have the authors described where all data underlying the findings will be made available when the study is complete?

Reviewer #1: Yes

Reviewer #2: No

5. Is the manuscript presented in an intelligible fashion and written in standard English?

Reviewer #1: Yes

Reviewer #2: Yes

6. Review Comments to the Author

You may also provide optional suggestions and comments to authors that they might find helpful in planning their study.

Reviewer #1: No additional comments. The authors have explained clearly how they revised the manuscript according to the points made by reviewer 2.

Reviewer #2: I appreciate the efforts to address my comments. Nevertheless, most of them have been overlooked and minor changes have been added to the main text.

As I mentioned before, models' assumptions must be carefully checked. This is true for the log-binomial model (rarely used in the statistical literature, as risk ratios cannot be calculated in retrospective (case-control) studies, where the total number of exposed people is not available), for which a comparison with more classical approaches is mandatory, but even more for the linear model for which the Gauss Markov assumptions must be checked. Please, provide a detailed residual analysis.

Similarly, the AIC can be used as guidance for model selection, but not to evaluate the goodness-of-fit of the model. Moreover, I am interested in the predictive ability of the log-binomial model. How well does the model recover the observed events?

Random effects modelling should be added and results compared.

7. PLOS authors have the option to publish the peer review history of their article (what does this mean?). If published, this will include your full peer review and any attached files.

Reviewer #1: No

Reviewer #2: No

---

## [Author Response · Author response to Decision Letter 1]

11 Apr 2024

11 April, 2024

PLOS ONE

Subject: Study Protocol Revisions – Response to Reviewers (2)

Dear Editorial Board of PLOS ONE and respected reviewers,

We extend our sincere appreciation for your thorough evaluation and feedback on our manuscript. For this second rebuttal, we sought guidance from Steven Teerenstra, statistician at the Radboudumc renowned for his expertise in stepped wedge design. 

To the best of our best knowledge, we have diligently addressed the comments provided by reviewer 2. 

We would like to express our profound gratitude to the entire Editorial Board and both reviewers for their valuable insights and guidance, which have contributed to the refinement of our manuscript. 

Kind regards, 

Luuk D. Drager

MD, MSc, PhD Candidate 

Radboud University Medical Center

Department of Surgery, route 618 

Postbus 9101, 6500 HB Nijmegen, the Netherlands

Email: luuk.drager@radboudumc.nl

Reviewer 1

“No additional comments. The authors have explained clearly how they revised the manuscript according to the points made by reviewer 2.”

Response

No additional revisions made. 

Reviewer 2

“I appreciate the efforts to address my comments. Nevertheless, most of them have been overlooked and minor changes have been added to the main text.

As I mentioned before, models' assumptions must be carefully checked. This is true for the log-binomial model (rarely used in the statistical literature, as risk ratios cannot be calculated in retrospective (case-control) studies, where the total number of exposed people is not available), for which a comparison with more classical approaches is mandatory, but even more for the linear model for which the Gauss Markov assumptions must be checked. Please, provide a detailed residual analysis.

Similarly, the AIC can be used as guidance for model selection, but not to evaluate the goodness-of-fit of the model. Moreover, I am interested in the predictive ability of the log-binomial model. How well does the model recover the observed events?

Random effects modelling should be added and results compared.”

Response

Firstly, we want to underscore one of the selected effect parameters, the risk ratio. We opted for the risk ratio as the effect parameter due to its superior interpretability compared to the odds ratio. Unlike the retrospective (case-control) studies referenced, all our data are prospectively collected as detailed in our manuscript. This also accounts for the data regarding postoperative complications in the historical cohort, since this is extracted from electronic patient files. Furthermore, the risk ratio aligns with the comparison within clusters inherent to the stepped wedge design. 

The prespecification of the primary analysis is crucial for mitigating data-dredging risks. Therefore, we will meticulously report the outcome of the prespecified analysis and provide additional analyses as supplementary information. Our selection of the primary analysis was based on simulation studies comparing various models, including random effect modeling with Poisson-error distribution, random effects modeling with binomial-error distribution, and fixed effects modeling with binomial-error distribution. After considering the expected cluster sizes, the fixed effects model with binomial error distribution emerged as the optimal choice, exhibiting superior power and type I error control.

We acknowledge the importance of model assessment, and thus, we intend to evaluate the adequacy of our model by examining whether its predictions systematically over- or underestimate observed data. This evaluation will involve assessing observed versus predicted plots and residual versus predicted plots. Specifically, we will ensure that the distribution of residuals along the predicted axis does not suggest any deviations from normality. Furthermore, we remain open to supplementing our primary analysis model with additional analyses, such as incorporating transformed outcomes, introducing or transforming explanatory variables, or exploring alternative models and link functions to enhance the model fit. We appreciate your suggestion regarding the utilization of a random effect model, and we added this as a supportive analysis, as mentioned in our revised manuscript on page 16 and page 17.

Within models of the same category, the AIC serves as a useful guide for making specific decisions. It is important to note that the log-likelihood of models must be calculated consistently to ensure uniformity; otherwise, the normalizing constant in the AIC would vary. When comparing models across different categories, we will primarily assess the discrepancy between predicted and observed data (prediction error magnitude). 

Furthermore, our main focus will be on evaluating the fit at the cluster level. Given that this is a cluster trial, the central inquire revolves around whether the overall outcome of clusters exhibits improvement.

---

## [Decision Letter · Decision Letter 2]

2 May 2024

Multimodal Prehabilitation (Fit4Surgery) in High-impact Surgery to Enhance Surgical Outcomes: Study Protocol of F4S PREHAB, a Single Center Stepped Wedge Trial

PONE-D-24-01585R2

Dear Dr. Luuk David Drager,

We’re pleased to inform you that your manuscript has been judged scientifically suitable for publication and will be formally accepted for publication once it meets all outstanding technical requirements.

Kind regards,

Stefano Turi

Academic Editor

PLOS ONE

Reviewers' comments:

Reviewer's Responses to Questions

**Comments to the Author**

1. Does the manuscript provide a valid rationale for the proposed study, with clearly identified and justified research questions?

Reviewer #1: Yes

Reviewer #2: Yes

2. Is the protocol technically sound and planned in a manner that will lead to a meaningful outcome and allow testing the stated hypotheses?

Reviewer #1: Yes

Reviewer #2: Yes

3. Is the methodology feasible and described in sufficient detail to allow the work to be replicable?

Reviewer #1: Yes

Reviewer #2: Yes

4. Have the authors described where all data underlying the findings will be made available when the study is complete?

Reviewer #1: Yes

Reviewer #2: Yes

5. Is the manuscript presented in an intelligible fashion and written in standard English?

Reviewer #1: Yes

Reviewer #2: Yes

6. Review Comments to the Author

You may also provide optional suggestions and comments to authors that they might find helpful in planning their study.

Reviewer #1: No additional comments,

Reviewer #2: Thank you very much for the replies to my comments. Though I do not fully agree with some of the statements, I recognize the efforts to properly account for my suggestions.

7. PLOS authors have the option to publish the peer review history of their article (what does this mean?). If published, this will include your full peer review and any attached files.

Reviewer #1: **Yes: **Ulla N Joensen

Reviewer #2: No

---

## [Editor Report · Acceptance letter]

10 May 2024

PONE-D-24-01585R2 

PLOS ONE

Dear Dr. Drager, 

I'm pleased to inform you that your manuscript has been deemed suitable for publication in PLOS ONE. Congratulations! Your manuscript is now being handed over to our production team.

Kind regards, 

on behalf of

Dr. Stefano Turi 

Academic Editor

PLOS ONE